# LANGUAGE-AGNOSTIC REPRESENTATION LEARNING OF SOURCE CODE FROM STRUCTURE AND CONTEXT

**Daniel Zügner, Tobias Kirschstein**
Technical University of Munich
{zuegnerd,kirschto}@in.tum.de

**Michele Catasta**
Stanford University
pirroh@cs.stanford.edu

**Jure Leskovec**
Stanford University
jure@cs.stanford.edu

**Stephan Günnemann**
Technical University of Munich
guennemann@in.tum.de

## ABSTRACT

Source code (*Context*) and its parsed abstract syntax tree (AST; *Structure*) are two complementary representations of the same computer program. Traditionally, designers of machine learning models have relied predominantly either on Structure or Context. We propose a new model, which jointly learns on Context and Structure of source code. In contrast to previous approaches, our model uses only language-agnostic features, i.e., source code and features that can be computed directly from the AST. Besides obtaining state-of-the-art on monolingual code summarization on all five programming languages considered in this work, we propose the first *multilingual* code summarization model. We show that jointly training on non-parallel data from multiple programming languages improves results on all individual languages, where the strongest gains are on low-resource languages. Remarkably, multilingual training only from Context does not lead to the same improvements, highlighting the benefits of combining Structure and Context for representation learning on code.

## 1 INTRODUCTION

Machine learning for code is an active and growing area of research which aims at building models that can learn semantically meaningful representations of programs. These embeddings can be used on downstream tasks, such as code generation, bug detection, or code summarization. We focus our work on two complementary data representations of programs: the source code (referred to as *Context* in this work), and the abstract syntax tree (AST; referred to as *Structure*). Traditionally, researchers and practitioners have decided to predominantly leverage either Structure or Context in their machine learning models. In this work, we show that jointly learning on Context and Structure improves representation learning on source code (see Fig. 1).

The source code representation naturally lends itself to models from natural language processing (NLP), e.g., long short-term memory networks (Hochreiter & Schmidhuber, 1997) (LSTM) or Transformers (Vaswani et al., 2017; Radford et al., 2019; Dai et al., 2019; Yang et al., 2019; Shaw et al., 2018). On the other hand, models leveraging the structure representations are typically based on graph neural networks (GNNs) (Kipf & Welling, 2017; Xu et al., 2019; Veličković et al., 2018; You et al., 2019; Hamilton et al., 2017; Li et al., 2015; Klicpera et al., 2020). While the AST representation makes the highly structured nature of source code explicit to the models, since most GNNs use the message-passing framework, their learned representations are inherently local and struggle to leverage long-range interactions.

Recently, Hellendoorn et al. (2020) have explored models that can leverage several representations, including both Structure and Context. Their Graph Relational Embedding Attention Transformer (GREAT) extends Shaw et al. (2018), which biases the self-attention computation in a localized way given the underlying graph. The language-specific representations used by GREAT include a combination of the data flow graph, control flow graph, syntactic edges (inspired by Allamanis et al. (2018)), etc. which require specialized pipelines and static analysis tools to be obtained.

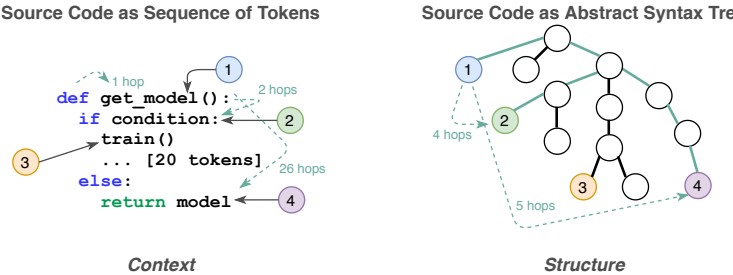

Figure 1: Context and Structure both encapsulate valuable information about source code. In this realistic example, token 1 and 4 are distant in the sequence of tokens (*Context*), but only 5 hops away when traversing the Abstract Syntax Tree (*Structure*). As such, a method that relies only on the sequence of tokens could neglect the relationship between a method name and its return variable. Conversely, token 1 and 2 showcase the opposite setting. Hence, unifying Structure and Context leads to a more powerful representation of source code.

We propose the CODE TRANSFORMER[1], which combines distances computed on Structure and Context in the self-attention operation. In contrast to the localized treatment via edges described above, we make the full Structure accessible to the model at each layer by computing pairwise distances on the AST, such as shortest path lengths. To this end, we draw inspiration from the XLNet architecture (Yang et al., 2019), which uses relative distances instead of absolute positions in the attention computation. Importantly, all our features are *language-agnostic*[2], i.e., can easily be computed for any programming language based on the source code and AST.

We use two datasets comprising 5 different programming languages in total, and evaluate the representations learned by our model on the task of code summarization, where the model predicts a method's name based on its body. Besides setting the state-of-the-art on all five languages for single-language training, we also train the first *multilingual* model for code summarization. This is enabled by the fact that our model uses only language-agnostic features that can easily be obtained for any programming language. Remarkably, training our model on multiple programming languages substantially improves the performance on all languages. Moreover, multilingual training only from Context does not lead to the same improvements, highlighting the benefits of combining Structure and Context for representation learning on code.

## 2  RELATED WORK

**Machine Learning for Code.** Early research learned language models on raw text data, e.g., (Wang et al., 2016; Raychev et al., 2014; Dam et al., 2016), providing evidence for the *naturalness assumption* (Hindle et al., 2012). For example, Allamanis et al. (2015) learned distributed representations of variables and methods, finding that they were indeed able to encode common semantic properties from the regularities present in source code. Alon et al. (2019b) also found evidence of semantic arithmetic in their embedding space, dubbed code2vec. These representations—and their variants like (Mou et al., 2016)—can then be used to predict sequences of identifier sub-tokens (Allamanis et al., 2015) or API calls (Acharya et al., 2007; Nguyen et al., 2017). They can be used as advanced auto-completion tools (Hindle et al., 2012; Bhoopchand et al., 2016), including for user-provided tokens like Variable Names (Raychev et al., 2014; Allamanis et al., 2014). These are useful for deobfuscating Android applications (Bichsel et al., 2016) for example.

Several works leverage structured graphical models for probabilistic models of source code, usually through parse trees (Maddison & Tarlow, 2014; Bielik et al., 2016). Unlike previous works where hand-crafted features were used as node features (Raychev et al., 2014) or as explicit semantic edges (Allamanis et al., 2018), our work does not augment the existing syntactic relationships between the

---

[1]Code at `www.daml.in.tum.de/code-transformer`, demo at `code-transformer.org`.

[2]We use the term language-agnostic to highlight that our model does not rely on language-specific features (e.g., program analysis edges), thus facilitating multi-language training, as it is possible to generate unified AST representations for different programming languages.

different elements to enhance the predictive capabilities of the model. Other approaches (Alon et al., 2018; Li et al., 2017) also leverage the AST structure, but linearize the graph by first traversing it.

**Learning representations of structured languages.** While models of language have dramatically improved in their ability to learn structure (syntax) and semantics from scratch, it can be argued that directly providing the model with the underlying structure of the language can help with generalization (Battaglia et al., 2018), managing long-ranging dependencies (Tai et al., 2015), or representing the compositional aspect of natural language (Socher et al., 2013). Notably, tree structures have shown promising results and inspired new architectures (Shen et al., 2019), including in the domain of source code (Fernandes et al., 2019), where the underlying syntax is directly available. Our work pursues this line of research, showing the benefits of explicitly integrating structural information as an inductive bias. Shiv & Quirk (2019) propose positional encodings for nodes on trees; however, their approach assumes regular trees, which is an unrealistic assumption when working with Abstract Syntax Trees, as an AST node can have arbitrarily many children, e.g., the arguments of a function.

**Graph Neural Networks.** GNNs provide a powerful tool for machine learning on graphs, thanks to their ability to recursively incorporate information from neighboring nodes in the network (Battaglia et al., 2018), naturally capturing the graph structure simultaneously with the nodes' features. (Gori et al., 2005; Scarselli et al., 2008) are able to learn vector representations of nodes and graphs in an end-to-end fashion, encoding structural and feature information in the embedding space. Under this model, GNNs have achieved state-of-the-art performance across a variety of tasks, such as node classification (Kipf & Welling, 2017; Hamilton et al., 2017; Klicpera et al., 2019a), link prediction (Zhang & Chen, 2018; Schlichtkrull et al., 2018), graph clustering (Defferrard et al., 2016; Ying et al., 2018) or graph classification (Ying et al., 2018; Dai et al., 2016; Duvenaud et al., 2015).

## 3 INTEGRATING STRUCTURE AND CONTEXT IN THE CODE TRANSFORMER

Self-attention is the core operation powering the Transformer. It enables the model to selectively focus on relevant parts of the input. The matrix form equation for attention with a single head is

$$\text{Attention}(\boldsymbol{Q}, \boldsymbol{K}, \boldsymbol{V}) = \text{softmax}\left(\frac{\boldsymbol{Q}\boldsymbol{K}^T}{\sqrt{d_k}}\right)\boldsymbol{V}, \tag{1}$$

where $\boldsymbol{Q}, \boldsymbol{K} \in \mathbb{R}^{N \times d_k}$ and $\boldsymbol{V} \in \mathbb{R}^{N \times d_v}$. $N$ is the number of input tokens, $d_k$ the key dimension, and $d_v$ the value dimension (typically we have $d_k = d_v$). The attention score of query $\boldsymbol{Q}_i$ and key $\boldsymbol{K}_j$ before softmax is

$$\boldsymbol{A}_{ij} = \boldsymbol{Q}_i^T \boldsymbol{K}_j = \boldsymbol{E}_i^T \boldsymbol{W}_q^T \boldsymbol{W}_k \boldsymbol{E}_j, \tag{2}$$

where $\boldsymbol{E}_i, \boldsymbol{E}_j \in \mathbb{R}^d$ are the $d$-dimensional embeddings of tokens $i$ and $j$, and $\boldsymbol{W}_q, \boldsymbol{W}_k \in \mathbb{R}^{d_k \times d}$ are the query and key projection matrices, respectively.

Observe that Eq. (2) contains no assumption about potential structure in the input domain: in the attention operation we compute *all* dot products of query and key vectors equally, effectively viewing them as unordered sets of vectors. This means, however, that the model is oblivious to structured inputs (such as text or graphs) and therefore is unable to distinguish, for example, a variable name occurring as an argument and in the return statement of a method.

In NLP, it is common to bias Transformers towards sequential inputs by adding positional encodings to the token embeddings. These positional encodings are obtained by applying an encoding function $\phi : \mathbb{R} \rightarrow \mathbb{R}^d$ to each token's position $p_i$. These positional encodings make the information about the sequence of tokens available to the model. Eq. (2) becomes:

$$\boldsymbol{A}_{ij} = (\boldsymbol{E}_i + \phi(p_i))^T \boldsymbol{W}_q^T \boldsymbol{W}_k (\boldsymbol{E}_j + \phi(p_j)),, \tag{3}$$

which factorizes into

$$\boldsymbol{A}_{ij} = \underbrace{\boldsymbol{E}_i^T \boldsymbol{W}_q^T \boldsymbol{W}_k \boldsymbol{E}_j}_{\text{(a) } \boldsymbol{A}_{ij}^{\text{cc}}} + \underbrace{\boldsymbol{E}_i^T \boldsymbol{W}_q^T \boldsymbol{W}_k \phi(p_j)}_{\text{(b) } \boldsymbol{A}_{ij}^{\text{cp}}} + \underbrace{\phi(p_i)^T \boldsymbol{W}_q^T \boldsymbol{W}_k \boldsymbol{E}_j}_{\text{(c) } \boldsymbol{A}_{ij}^{\text{pc}}} + \underbrace{\phi(p_i)^T \boldsymbol{W}_q^T \boldsymbol{W}_k \phi(p_j)}_{\text{(d) } \boldsymbol{A}_{ij}^{\text{pp}}}. \tag{4}$$

We can interpret the terms (a)-(d) as follows. (a) $\boldsymbol{A}_{ij}^{\text{cc}}$ is the contribution from the 'match' between the content embeddings of tokens $i$ and $j$; (b) $\boldsymbol{A}_{ij}^{\text{cp}}$ steers the attention towards certain positions based

on the content of token $i$; (c) $\boldsymbol{A}_{ij}^{\mathrm{pc}}$ biases towards content embeddings based on the position of token $i$; (d) $\boldsymbol{A}_{ij}^{\mathrm{pp}}$ controls which positions should attend to which other positions.

In our model, we adopt the formulation of Dai et al. (2019); Yang et al. (2019). They modify Eq. (4) by replacing the *absolute* position encodings $\phi(p_i)$ with *relative* position encodings $\phi(r_{i \to j})$:

$$\boldsymbol{A}_{ij}^{\mathrm{rel}} = \boldsymbol{E}_i^T \boldsymbol{W}_q^T \boldsymbol{W}_k \boldsymbol{E}_j + \boldsymbol{E}_i^T \boldsymbol{W}_q^T \boldsymbol{W}_r \phi(r_{i \to j}) + \boldsymbol{u}^T \boldsymbol{W}_k \boldsymbol{E}_j + \boldsymbol{v}^T \boldsymbol{W}_r \phi(r_{i \to j}), \quad (5)$$

where $r_{i \to j}$ is the relative distance from token $i$ to token $j$ in the sequence, $\boldsymbol{u}, \boldsymbol{v} \in \mathbb{R}^{d_k}$ are learnable bias vectors, and $\boldsymbol{W}_r$ is a key projection matrix for the relative distances. Besides fixing issues with absolute position encodings such as ambiguity when processing two sentences at a time, Eq. (5) enables native application of the powerful self-attention operation on domains such as graphs, where absolute coordinates are not available. We adopt the (non-trainable) sinusoidal encoding function proposed by (Vaswani et al., 2017) for all relations; see Appendix A.1 for details on the distance encoding function.

### 3.1 INTEGRATING SOURCE CODE AND AST REPRESENTATIONS OF PROGRAMS.

To enable the model to integrate information both the Context and Structure of programs, we modify Eq. (5) to be able to incorporate multiple different relations. To this end, we use one key projection matrix $\boldsymbol{W}_r^{(s)}$ *per relation* $s$, and sum their contributions in the raw attention score. This enables the CODE TRANSFORMER to combine information from multiple relations between tokens in the attention computation. Besides the token distance in the Context, we include pairwise relations based on the AST as described in the following. See Fig. 2 for a visualization of the Structure distances we use.

**Shortest path length.** We include the number of hops required to reach node $j$ starting from node $i$ and vice versa. Here, we treat the AST as an undirected graph, since otherwise most distances would be undefined: e.g., all other nodes in the AST would be unreachable from the leaves.

Similar to the distance of two tokens on the source code sequence, the shortest-path length is a global distance. This makes the whole graph structure acces-

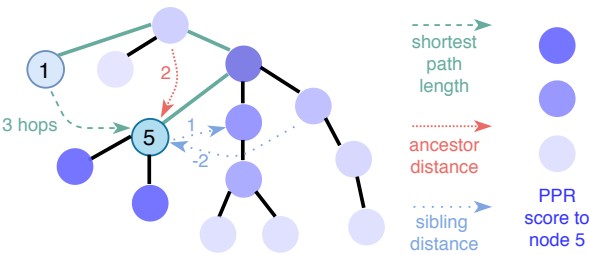

Figure 2: Structure distances used by our model.

sible to the model at each layer. In contrast, Hellendoorn et al. (2020) add bias terms to the attention computation only for edges (i.e. shortest-path distance of 1), which is a local operation that only exchanges information between immediate neighbors (similar to message passing in GNNs). The equivalent localized operation on the source code sequence would be to treat the sequence as a chain graph and only compute attention terms for neighboring tokens, which in turn highlights the benefit of non-local attention operations.

**Ancestor distance.** Since we treat the ASTs as undirected for the computation of the shortest-path length, we lose the direction information of the edges. To avoid this, we also include the distance on the ordered set of ancestors and descendants of a node in the AST (red arrow in Fig. 2). Again, we include number of (vertical) hops to avoid locality in the attention computation. For example, a node $r_{i \to j} = 2$ for "grand-children" $j$ of $i$, and $r_{j \to i} = -2$ in the other direction.

**Sibling distance.** The neighbor sets in graphs are typically considered to be unordered, but in an AST, the order of children encodes their order of occurrence in the source code. To avoid information loss when encoding the AST, we further include the distance on the ordered set of siblings $\{v_i\}$ of a node, where we again avoid locality by encoding the number of hops, i.e. $r_{v_1 \to v_3} = 2$ and $r_{v_3 \to v_1} = -2$.

**Personalized PageRank (Page et al., 1999) (PPR).** PPR is a well-studied proximity measure which has been shown to be very effective in learning with graphs (Klicpera et al., 2019a;b; Bojchevski et al., 2020). PPR captures the local graph structure around a pair of nodes $(i, j)$. E.g., if $i$ has

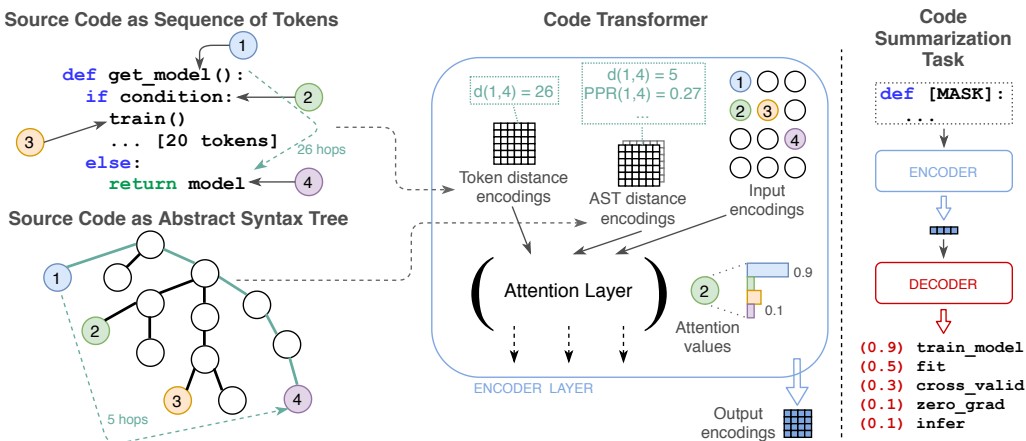

Figure 3: **Left**: Sequence (Context) and AST (Structure) representation of an input code snippet. **Center**: The CODE TRANSFORMER jointly leverages the sequence of tokens and the Abstract Syntax Tree to learn expressive representations of source code. In addition to the input token and node embeddings the model uses different distances between the tokens, e.g., shortest paths on the AST or personalized PageRank, to reason about their relative positions. The output embeddings can be used for downstream tasks such as code summarization (**right**).

many neighbors, its PPR score for $j$ will be low even when they are only few hops apart, which complements the purely hop-based distances described above.

**Input embeddings to the model.** To combine the Context and Structure information, we assign each token in the sequence to an AST node by selecting the AST node whose range in the source code is the shortest one containing the token. We concatenate the (sub-) token embeddings with the embedding of the token's assigned AST node type as well as the token type returned by the tokenizer. That is, among all the internal nodes, we use as input only those corresponding to a token in the sequence; however, the remaining internal nodes can used by the model since their presence affects the distances between the remaining AST nodes. See Appendices A.3 and A.4 for details.

## 3.2 EFFICIENT RELATIVE ATTENTION COMPUTATION.

Naïvely, we need to compute and materialize a tensor of dimension $N \times N \times d$ to hold all pairwise relative position encodings $\phi(r_{i \to j})$ in Eq. (5) , where $N$ is the input length. This is prohibitive for fast GPU training. While for discrete distance values (e.g., sequence distance or shortest-path length on a graph) we only need to compute unique distance values occurring in the input, this does not generalize to continuous distances such as PPR. Therefore, we propose a constant-time approximation of the relational attention computation by grouping the values into $k \ll N^2$ bins. Since closer samples are typically more relevant for a query sample, we increase the bin widths exponentially with growing distance values. Throughout our experiments we have found the CODE TRANSFORMER to be relatively insensitive to the number of bins; we thus set $k = 32$ in our experiments.

## 4 EXPERIMENTAL SETUP

Code summarization is one of the most popular tasks in machine learning for code. Given the body of a function, the task is to predict the function's name. As observed by Alon et al. (2019b) and Allamanis et al. (2016), this is a useful benchmark as method names in open-source projects tend to be precise and descriptive, and functions typically form complete logical units. See Fig. 3 (right) for a visual overview of the task. We use two complementary representations of programs: the source code as a sequence of tokens (Context) and the AST (Structure). As shown in Fig. 3 (left), tokens that are far away on the sequence may be very close on the AST and vice versa. In this task we make use of the CODE TRANSFORMER's ability jointly leverage both Structure and Context and show that

it improves learning. Further, we show the benefit of using only language-agnostic features in our model by training the first multilingual model for code summarization.

**Datasets.** To highlight the benefit of only relying on language-agnostic representations such as source code and abstract syntax trees, we evaluate on challenging datasets in four programming languages introduced in the CodeSearchNet (CSN) Challenge (Husain et al., 2019): Python, Javascript, Go, and Ruby. Similar to Java-small, the datasets from CodeSearchNet have been carefully deduplicated by the creators to avoid data leakage from the training set, e.g., via copy-and-paste code.

We further evaluate on Java-small (Allamanis et al., 2016), a popular and challenging code summarization dataset. It contains 11 open-source Java projects. We use the split as in Alon et al. (2019a), where 9 of these projects are used for training, one for validation, and one for test. The dataset contains roughly 700K samples (function definitions). Moreover, we also experiment with pre-training our model on Java-medium and Java-large (Alon et al., 2019a) before fine-tuning on Java-small, making sure to avoid leakage by removing the test and validation projects of Java-small from the pre-training dataset. See Table 1 for a summary of the datasets we use in this work.

| Dataset | Samples per partition | | |
| | Train | Val. | Test |
| --- | --- | --- | --- |
| CSN-Python | 412,178 | 23,107 | 22,176 |
| CSN-Javascript | 123,889 | 8,253 | 6,483 |
| CSN-Ruby | 48,791 | 2,209 | 2,279 |
| CSN-Go | 317,832 | 14,242 | 14,291 |
| Java-small | 691,974 | 23,844 | 57,088 |

Table 1: Dataset statistics.

**Preprocessing.** Each token of the source code is split into subtokens respective to code naming conventions, i.e., `get_TrainingData` is converted to `[get, training, data]`. Following Alon et al. (2019a) we use at most six subtokens for the method names, truncating longer function names if necessary. In addition to the tokenized source code we produce an AST for each method using the open-source AST parser Semantic[3]. We limit the vocabulary to subtokens with at least 100 occurrences in the training set, and only consider snippets with 512 or fewer tokens (after removing punctuation). We refer the reader to the appendix for further details on the data preprocessing.

**Pointer network.** We add a pointer network (Vinyals et al., 2015) (as described in Fernandes et al. (2019)) to the decoders of all Transformer-based models. This enables them to enhance their predictions by pointing at positions in the input sequence. For instance, when predicting the method name `get_url`, the model can point directly to occurrences of the variable `url`. This often improves results for less frequent tokens, and even enables the model to predict tokens which are not in the vocabulary by pointing at their positions in the input.

**Baselines.** We compare with code2seq (Alon et al., 2019a), the Graph Relational Embedding Attention Transformer (GREAT) (Hellendoorn et al., 2020), and the BiLSTM+GNN→LSTM+Pointer model presented in Fernandes et al. (2019). Code2seq is a non-Transformer model and state of the art for code summarization using only AST information. GREAT is a recent Transformer model using the framework presented in (Shaw et al., 2018) to bias the attention via edges. In the original formulation, GREAT additionally uses hand-crafted, language-specific edges such as dataflow, 'computed from', or 'next lexical use' edges, which require specialized preprocessing and static analysis tools. While this approach of leveraging language-specific features can certainly improve results on specific tasks and programming languages, our goal is to have a flexible model that can be used on any programming language. Since the specialized preprocessing used by GREAT is proprietary and not public, we produce the results for GREAT using edges from the AST instead, i.e. it has access to the same information as our proposed model. Note that the preprocessing of Fernandes et al. (2019) is language specific, which is why we only compare with their results on Java-small.

## 5 RESULTS

### 5.1 MONOLINGUAL CODE SUMMARIZATION

**CSN dataset.** First, we study the performance (measured by F1 score) of our model and the baselines on the traditional setting, where training and evaluation are performed on a single programming language. The results are shown in the upper part of Table 2. The CODE TRANSFORMER (without

---

[3]`https://github.com/github/semantic`

| Model | Python | | | Javascript | | | Ruby | | | Go | | |
|---|---|---|---|---|---|---|---|---|---|---|---|---|
| | Prec. | Rec. | F1 | Prec. | Rec. | F1 | Prec. | Rec. | F1 | Prec. | Rec. | F1 |
| code2seq | 35.79 | 24.85 | 29.34 | 30.18 | 19.88 | 23.97 | 23.23 | 10.31 | 14.28 | 52.30 | 43.43 | 47.45 |
| GREAT | 35.07 | 31.59 | 33.24 | 31.20 | 26.84 | 28.86 | 24.64 | 22.23 | 23.38 | 50.01 | 46.51 | 48.20 |
| Ours w/o structure | 37.38 | 31.98 | 34.47 | 33.17 | 26.70 | 29.59 | 29.85 | 25.87 | 27.72 | 51.78 | 47.57 | 49.59 |
| Ours w/o pointer net | 37.74 | 31.85 | 34.55 | 33.12 | 28.70 | 30.75 | 23.32 | 25.21 | 24.23 | 54.31 | 50.12 | 52.13 |
| Ours | 36.40 | 33.66 | 34.97 | 35.06 | 29.61 | 32.11 | 31.42 | 24.46 | 27.50 | 55.10 | 48.05 | 51.34 |
| code2seq (Multilanguage) | 34.49 | 25.49 | 29.32 | 31.62 | 22.16 | 26.06 | 23.97 | 17.06 | 19.93 | 52.70 | 44.36 | 48.17 |
| GREAT (Multilanguage) | 36.75 | 31.54 | 33.94 | 33.58 | 27.78 | 30.41 | 30.05 | 24.33 | 26.89 | 52.65 | 48.30 | 50.38 |
| Ours w/o structure (Mult.) | 38.48 | 30.14 | 33.80 | 35.38 | 27.41 | 30.89 | 32.61 | 26.76 | 29.40 | 55.03 | 47.34 | 50.90 |
| Ours w/o pointer (Mult.) | 38.91 | 33.12 | 35.78 | 37.21 | 29.75 | 33.07 | 34.52 | 27.31 | 30.50 | 56.07 | 50.76 | 53.28 |
| Ours (Multilanguage) | 38.89 | 33.82 | 36.18 | 36.95 | 29.98 | 33.10 | 33.93 | 28.94 | 31.24 | 56.00 | 50.44 | 53.07 |
| Ours (Mult. + Finetune) | 39.85 | 32.79 | 35.98 | 37.00 | 29.79 | 33.00 | 35.85 | 27.75 | 31.28 | 55.63 | 51.12 | 53.28 |
| Ours (Mult. + LM Pretrain) | 39.67 | 35.29 | 37.35 | 37.06 | 31.94 | 34.31 | 35.19 | 29.36 | 32.01 | 57.73 | 51.89 | 54.65 |

Table 2: Code summarization results on the CSN dataset (micro F1).

multi-language training) substantially outperforms all other models on all but one language, highlighting the effectiveness of jointly learning from Structure and Context. The only exception is Ruby, where it performs on par with its Context-only variant. We attribute this to the fact that there are relatively few samples in the Ruby dataset, and that Ruby is an dynamically typed language, which could make the Structure less powerful for learning. Interestingly, the Context-only CODE TRANS-FORMER outperforms GREAT on all languages. We attribute this to the fact that GREAT uses the Structure of the programs only in a localized way (see Sec. 3.1). Another noteworthy finding is that code2seq performs comparably to the Transformer-based baselines on Go. We hypothesize that ASTs are more informative on Go since it is a compiled and strongly typed language.

**Java-small results.** In Table 3 we present code summarization results on the Java-small dataset. Among all models equipped with a pointer network, the CODE TRANSFORMER (without pretraining) obtains state-of-the-art on code summarization, outperforming all baselines, including the previous state-of-the-art on Java-small proposed by Fernandes et al. (2019). Further, pre-training on Java-medium and Java-large on the permutation language modeling objective (Yang et al., 2019) substantially improves precision, recall, and F1 score after fine-tuning on Java-small. To avoid leakage, we exclude the projects used in the validation and test splits of Java-small from pre-training.

**Ablation study.** We further perform ablations where we remove our model's access to the Context or Structure, also presented in Table 3.

*With pointer network.* We find that both ablations lead to a substantial drop in performance, highlighting the benefit of learning jointly from Structure and Context. Interestingly, the model without access to the Structure performs slightly better than the variant without Context. Note that our model without Structure is related to the XLNet (Yang et al., 2019) model, where we add a pointer network to the decoder and concatenate the token types to their respective input tokens (see Appendix A.4). *Without pointer network.* We repeat the ablation on the variants without pointer network. Here, the variant without Context performs better than the variant without Structure,

| Model | Prec. | Rec. | F1 |
|---|---|---|---|
| *Without pointer net* | | | |
| code2seq | 51.23 | 37.31 | 43.18 |
| Ours w/o structure | 50.70 | 45.49 | 47.96 |
| Ours w/o context | 51.81 | 46.04 | 48.75 |
| Ours | 50.33 | 46.80 | 48.50 |
| *With pointer net* | | | |
| Fernandes et al. (2019) | - | - | 51.4 |
| GREAT | 53.60 | 46.41 | 49.75 |
| Ours w/o structure | 55.48 | 46.07 | 50.34 |
| Ours w/o context | 54.45 | 45.29 | 49.45 |
| Ours | 54.85 | 49.84 | 52.22 |
| Ours + Pretrain | 57.02 | 50.87 | 53.77 |

Table 3: Results on Java-small and ablation study.

indicating that the pointer network helps to compensate for the lack of access to Structure. The Structure-only variant (w/o pointer net) of our model even outperforms the full variant in this scenario. Inspection of the results revealed that the Structure-only variant has better performance on longer method names, which have an outsize influence on the micro-F1 score used in this work.

*Ablation of the AST-based distances.* In Table 4 we compare the performance of our model when trained with each of the four different AST distances (sibling shortest paths, ancestor shortest paths, shortest paths, personalized PageRank; see Section 3.1). Here, the model is trained on Java-small in the Structure-only setting and without pointer network. For reference, we also show the results of training our model using all four AST distance functions (c.f. Table 3). We find that, while the

personalized PageRank distance performs best on its own, each of the individual distances on their own performs substantially worse than their combination, highlighting the usefulness of combining the distances in our model as well as their complementary nature.

## 5.2 MULTILINGUAL CODE SUMMARIZATION

**Setup.** A key contribution of our proposed architecture is that it only uses language-agnostic features, i.e. the source code and features that can be directly computed from the AST. We use this fact to study the first *multi-language* code summarization model. We train our model jointly on Python, Javascript, Ruby, and Go. The shared sub-token vocabulary is the union of the individual vocabularies, enabling us to evaluate the multi-language model

| AST distance | F1 score |
|---|---|
| Sibling shortest paths | 46.17 |
| Ancestor shortest paths | 47.89 |
| Shortest paths | 47.76 |
| Personalized PageRank | 48.47 |
| All the above (c.f. Table 3) | **48.75** |

Table 4: AST distance ablation study.

on the individual languages and compare with the single-language models. As proposed by Conneau & Lample (2019), we add a learned language embedding to each input embedding.

**Results.** In the lower part of Table 2 we can see the results of training our CODE TRANSFORMER *jointly* on all four programming languages. Our multi-lingual variants substantially outperform the mono-lingual models on all languages. The strongest improvement is on Ruby, which is also the programming language with the smallest number of samples in the dataset. Fine-tuning on the individual languages after joint training on code summarization only has a marginal effect on performance, indicating that the multilingual objective is well-aligned with the individual languages. In the last row, we have a variant of our model where we pre-train on the multi-lingual masked language modeling task, followed by finetuning on code summarization on the individual languages.

Further, we observe that similar to the results on Java-small, removing the pointer network generally leads to weaker performance. One notable exception is Go, where the variant without the pointer network performs better in terms of F1 score. Our investigation revealed that there seems to be some violation of the i.i.d. assumption in the split provided by the creators of the dataset. In Figure 7 we show that in the test partition of the Go dataset, the share of tokens from the labels also occurring in the methods' bodies – exactly the scenario where the pointer network can improve predictions – is substantially lower compared to the train/validation partitions.

Remarkably, the multi-language Context-only variant (i.e. without access to the Structure) performs substantially worse than the full multi-language variant. This highlights that Structure is crucial to exploit the commonalities of different programming languages. Also notably, the GREAT baseline's results also improve substantially when trained in the multi-language setting, though it is still outperformed by our model. However, our results indicate that any representation learning model for code can benefit from multi-language training, especially when evaluating on low-resource languages.

In Table 16 we present results using the sample-F1 score. At the time of submission, our mono-lingual model on Python outperforms the state of the art on the `ogbg-code2`[4] (Hu et al., 2020) leaderboard by 112%, and our multilanguage variant with LM pretraining outperforms it by 122%.

**Qualitative analysis of multilingual representations.** Learning the CODE TRANSFORMER on multiple programming languages jointly provides us with embeddings in a shared representation space. In Fig. 4 we show a t-SNE (Maaten & Hinton, 2008) visualization of the ca. 40,000 snippets from the validation sets of four programming languages from the CSN dataset. For the embedding of a snippet, we use the representation of the method name in the final layer of the encoder. Note that the true method names are masked, i.e., inaccessible to the model. Further, note that in contrast to the monolingual embeddings learned by Kanade et al. (2020), the embeddings we evaluate are learned on the task of code summarization (though a similar study could be performed by using our model that was trained on the traditional language modeling pretraining task on multiple languages).

While snippets from the same language tend to be grouped together, there are interesting intersections of the different programming languages. For example, we highlight all methods whose names start with the subtoken `parse` or `main`. We see that snippets starting with `parse` are predominantly in an intersection region of Python and Javascript. From these snippets, we display the

---

[4]`https://ogb.stanford.edu/docs/graphprop/#ogbg-code2`

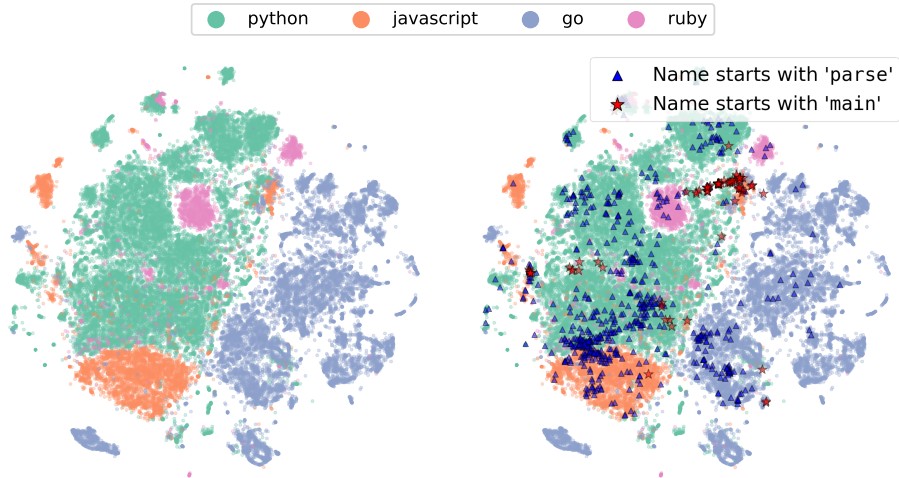

Figure 4: t-SNE visualization of the CODE TRANSFORMER's learned multilingual representations.

```python
def parseBool(s):
    l = s.lower()
    if l in ("true", "t", "1"):
        return True
    if l in ("false", "f", "0"):
        return False
    raise Exception(
        "Unable to convert string '%s'"
        "to a boolean value" % s
    )
```

```javascript
function jscoverage_getBooleanValue(s) {
    s = s.toLowerCase();
    if (s === 'false' || s === 'f'
        || s === 'no' || s === 'n'
        || s === 'off' || s === '0') {
        return false;
    }
    return true;
}
```

Figure 5: Example snippet starting with `parse` (left) and its best embedding match from other languages (right). Both methods parse an input string to convert it into a boolean value. Note that even though they are semantically very similar, their method names are not; nonetheless, their representations in the CODE TRANSFORMER encoder reflect their semantic similarity.

cross-language pair with smallest Euclidean embedding distance in Fig. 5. Remarkably, both snippets are effectively the same method in Javascript and Python – it is worth reminding that the model has never seen any parallel data during training. On the other hand, snippets starting with `main` tend to lie at an intersectional region of Python, Javascript, and Go. In Table 6 in the appendix we show additional cross-lingual pairs with similar embeddings, including a failure case of a `main` function, where embedding distance is not representative of semantic similarity. We attribute this to the fact that we used the encoder output embedding of the *masked method name* – the representation used by the decoder to predict the method name – as a snippet's representation. Thus, snippets with completely different semantics (as is to be expected for very generic method names starting with `main`) have similar representations because they are predictive of the method name.

As another qualitative insight into the representations learned by the CODE TRANSFORMER we have found that the language embeddings of languages with similar roots in language design are close; see Table 5 in the appendix for the pairwise similarity matrix of the learned language embeddings.

## 6 CONCLUSION

We present the CODE TRANSFORMER, which learns jointly from Structure and Context of programs while only relying on language-agnostic features. Our model obtains state-of-the-art performance on code summarization on five different programming languages. Besides these results for training on individual languages, the language-agnostic nature of our model allows us to train it *jointly* on multiple programming languages. The resulting multilingual model substantially outperforms its mono-lingual variant on all programming languages, setting the state of the art on each language. We observe the largest improvement from multilingual training on the language with fewest resources, indicating that multilingual training can improve learning for less widely used programming languages. Remarkably, multilingual training only from Context does not lead to the same improvements, highlighting the benefits of combining Structure and Context.

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

|  | Python | Javascript | Go | Ruby |
|---|---|---|---|---|
| Python | 1.00 | 0.43 | 0.42 | 0.79 |
| Javascript | 0.43 | 1.00 | 0.84 | 0.39 |
| Go | 0.43 | 0.84 | 1.00 | 0.38 |
| Ruby | 0.79 | 0.39 | 0.38 | 1.00 |

Table 5: Pairwise cosine similarities of the learned language embeddings of the CODE TRANSFORMER.

## ACKNOWLEDGEMENTS

We are grateful to Dylan Bourgeois for having paved the way to this research contribution with his thesis work (Bourgeois, 2019). We further thank Simon Geisler for his helpful suggestions and proofreading the paper, as well as the anonymous reviewers for their constructive feedback and fruitful discussions.

This research was supported by the TUM International Graduate School of Science and Engineering (IGSSE). Stanford University is supported by DARPA under Nos. N660011924033 (MCS); ARO under Nos. W911NF-16-1- 0342 (MURI), W911NF-16-1-0171 (DURIP); NSF under Nos. OAC-1835598 (CINES), OAC-1934578 (HDR), CCF-1918940 (Expeditions), IIS-2030477 (RAPID); Stanford Data Science Initiative, Wu Tsai Neurosciences Institute, Chan Zuckerberg Biohub, Amazon, JPMorgan Chase, Docomo, Hitachi, JD.com, KDDI, NVIDIA, Dell, Toshiba, Intel, and UnitedHealth Group. Jure Leskovec is a Chan Zuckerberg Biohub investigator.

## A  APPENDIX

### A.1  DISTANCE ENCODING FUNCTION

For encoding scalar relation values via vectors we employ encoding functions $\phi : \mathbb{R} \rightarrow \mathbb{R}^d$, where $d$ is the model's embedding dimension. We choose the popular sinusoidal encoding function presented in Vaswani et al. (2017):

$$\phi(r_{i \rightarrow j})_{2k} = \sin\left(\frac{r_{i \rightarrow j}}{M^{2k/d}}\right) \qquad \phi(r_{i \rightarrow j})_{2k+1} = \cos\left(\frac{r_{i \rightarrow j}}{M^{2k/d}}\right),$$

where $1 \leq k < d/2$ is the position in the encoding vector and $M$ is some constant; we adopt $M = 10,000$ as chosen by (Vaswani et al., 2017). Note that the distance encoding functions have no trainable parameters.

### A.2  MULTILINGUAL REPRESENTATION ANALYSIS

In Table 5, we show the pairwise cosine similarities of the learned language embeddings of the CODE TRANSFORMER. We can see that the pairs Python-Ruby and Javascript-Go have similar language embeddings. This aligns well with roots of language design and common use cases of the languages.

Moreover, in Table 6, we show selected snippets starting with `is`, `main`, or `load` (left) and their best embedding matches from other languages (right).

### A.3  DATA PREPROCESSING

#### A.3.1  TEXTUAL CODE SNIPPET PREPROCESSING

1. **Tokenize** code snippets with Pygments language-specific tokenizer.
2. **Remove comments** (both multi-line, single-line and doc comments). The comment token types. `pygments.token.Comment` and `pygments.token.Literal.String.Doc` that are generated by Pygments are used to identify comments.

```javascript
function _isEqualArray(a, b) {
    if (a === b) {
        return true;
    }
    if ((a === undefined) ||
        (b === undefined)) {
        return false;
    }
    var i = a.length;
    if (i !== b.length){
        return false;
    }
    while (i--) {
        if (a[i] !== b[i]) {
            return false;
        }
    }
    return true;
}
```

```go
func areSameFloat32Array(a, b []float32) bool {
  if len(a) != len(b) {
    return false
  }
  for i := 0; i < len(a); i++ {
    if a[i] != b[i] {
      return false
    }
  }
  return true
}
```

```javascript
function main() {
    var rawData = $('.HeaderTexture[data-login-
                user-email]').data();
    if (rawData) {
        me = {
            name: rawData.loginUserName,
            mail: rawData.loginUserEmail
        };
        getCartId(function (cart) {
            me.cart = cart;
            injectMenu();
            refreshUsers(actions.updateUsers);
            listenOnChanges(onChange);
            listenOnOrderConfirm(onConfirm);
        });
    } else {
        callback('no user');
    }
}
```

```go
func TaskSayHello(t *tasking.T) {
  username := t.Flags.String("name")
  if username == "" {
    user, _ := user.Current()
    username = user.Name
  }

  if t.Flags.Bool("verbose") {
    t.Logf("Hello %s, the time now is %s\n",
            username, time.Now())
  } else {
    t.Logf("Hello %s\n", username)
  }
}
```

```python
def _load_rule_file(self, filename):

    if not (os.path.exists(filename)):
        sys.stderr.write(
            "rflint: %s: No such file or "
            "directory\n" % filename
        )
        return
    try:
        basename = os.path.basename(filename)
        (name, ext) = os.path.splitext(basename)
        imp.load_source(name, filename)
    except Exception as e:
        sys.stderr.write(
            "rflint: %s: exception while "
            "loading: %s\n" % (filename, str(e))
        )
```

```go
func Backup(filename string) error {

  info, err := os.Stat(filename)
  if err != nil {
    if os.IsNotExist(err) {
      return nil
    }
    return err
  }
  if info.Size() == 0 {
    return nil
  }

  files, err := filepath.Glob(
    filename + _BACKUP_SUFFIX
  )
  if err != nil {
    return err
  }

  numBackup := byte(1)

  if len(files) != 0 {
    lastFile := files[len(files)-1]
    numBackup = lastFile[len(lastFile)-2] + 1
    if numBackup > '9' {
      numBackup = '1'
    }
  } else {
    numBackup = '1'
  }

  return Copy(filename,
            fmt.Sprintf("%s+%s~", filename,
                        string(numBackup)))
}
```

Table 6: Selected snippets starting with is, main, or load (left) and their best embedding matches from other languages (right).

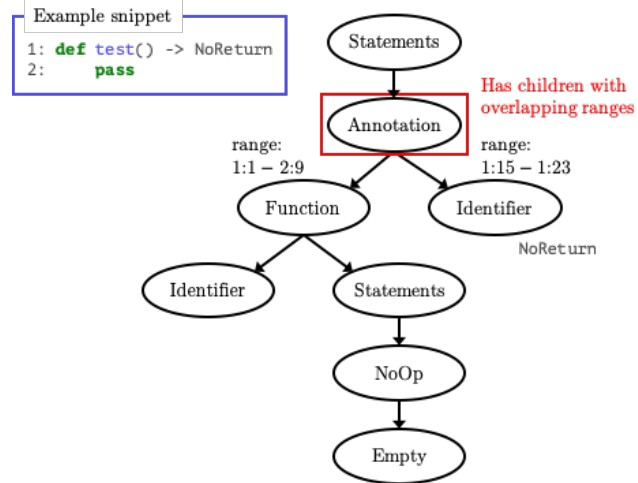

Figure 6: Example snippet and its corresponding AST obtained from GitHub Semantic.

3. **Empty lines** are removed.

4. **Hard coded strings and numbers** are replaced with a special [MASK_STRING] and [MASK_NUMBER] token.

5. **Indentation** style of the code snippet is detected and whitespace characters at the beginning of a line are replaced with a single [INDENT] or [DEDENT] token when indentation changes.

6. Tokens are further **split into sub tokens**, e.g., setBottomHeight → ['set', 'bottom', 'height']. Throughout our experiments, we use 5 input sub tokens. If a token consists of less than 5 sub tokens, the remaining spaces are filled with a special [PAD] token.

7. Any remaining tokens that only consist of **white spaces are removed**. The only white space characters that are kept are line breaks '\n'.

8. Any code snippets where the Pygments tokenizer **cannot parse a token** are **discarded**.

### A.3.2 STAGE 1 PREPROCESSING (GENERATION OF ASTS)

1. Stripped code snippets are used to **generate language-specific ASTs**. For Java, we use the AST parser from the **java-parser** project. The ASTs contain node types and source ranges. For Python, JavaScript, Ruby and Go, we use **semantic**.

2. Snippets that lead to an **AST parse error are discarded**.

3. We calculate a **mapping between tokens and nodes in the AST**. Every token is assigned to the node in the AST with shortest source range that still encompasses the source range of the token.

   To find such a node, we originally intended to make use of the assumption that source ranges of child nodes do not overlap. Then, one could easily find the node with smallest encompassing source range by greedily selecting at every layer in the AST the child that encompasses the token's source range (there can only be at most one child that fulfills this). However, this assumption does not hold for all ASTs (see Figure 6 for an example). As a heuristic, we greedily select the child node with the shorter source range in case there were multiple child nodes with encompassing source ranges. This approximation seems to be sufficient in our case, and limits runtime as we do not have to consider multiple paths in the AST. It is also sufficient to stop when no child node encompasses the source range of the token, as in ASTs the source ranges of child nodes are always contained in the source ranges of their parent.

### A.3.3 STAGE 2 PREPROCESSING (CALCULATION OF DISTANCE MATRICES)

1. Tokens are **vocabularized**. Any token occurring less than 100 times in the training set is **replaced by an `<unk>`** token.

2. We calculate multiple **pair-wise relations** between nodes in the AST:

   - Personalized Page Rank (PPR)
     We interpret the negative logarithm of PPR as a distance. We use a teleport probability of $\alpha = 0.15$ and a threshold of $e^{-5}$, i.e., anything with $-\log PPR > 5$ is considered unreachable
   - Shortest path length between two nodes
   - Ancestor shortest paths (bidirectional). That is, the parent has an ancestor shortest path distance of 1 to all its children and the child has a distance of -1 to its parents. We consider nodes that are not ancestors or descendants of a node (i.e. not reachable by following only parent or only child relations) as not connected in the ancestor shortest paths relation. We encode this with a very large value in their distance; we have found a value of $1,000$ to work well in practice.
   - Next sibling shortest paths (bidirectional, analogous to the ancestor shortest paths)

   Note that the ancestor shortest paths and next sibling shortest paths are required because treating the AST as a normal graph leads to ambiguity. In a graph, the neighbors of a node have no ordering; however in the AST, the order of the children of a node reflects their order in the code. Therefore, we explicitly include the next sibling shortest paths. The ancestor shortest paths would not be required if we treated the AST as a directed graph; in this case, however, a leaf node could not reach any other node in the AST, and therefore both PPR and shortest path length are not useful in this case. Therefore, we model the AST as undirected and inject the ancestor / child edges to avoid ambiguity.

3. **Distance values are binned** into 32 bins using area-based exponential binning with a growth factor of 1.3, i.e., the area of a bin's rectangle ($x$: bin range, $y$: number of values in bin) will be approximately 1.3 times bigger for the next bin (going away from the bin that contains the zero value). Additionally, for discrete distance measures (such as sequence distance or shortest path length), we hard-code 9 values around 0 to have their own bins. For instance, on the sequence distance the values $-4, -3, \ldots, 4$ have their individual bins, and around those values we employ the exponential binning.

4. Punctuation tokens (such as points or brackets) are removed from the input sequence, as experiments showed that their presence does not improve performance but slows down training due to bigger input sizes.

5. Snippets that are longer than `MAX_NUM_TOKENS` after punctuation tokens are removed are discarded from the training set. Throughout our experiments, we use `MAX_NUM_TOKENS = 512`. During evaluation on the test set, we use `MAX_NUM_TOKENS = 1000`.

## A.4 INPUT EMBEDDINGS TO THE MODEL

Besides its five subtokens (e.g., `['get', 'data', '[PAD]', '[PAD]', '[PAD]']`), each input token has a token type (coming from the Pygments tokenizer) and an AST node type. The AST node type is the type of the node assigned to each respective token, as described in Section A.3.2. We concatenate the embeddings of the five subtokens, the token type, and the AST node type. Then, we apply a linear layer (without activation function) to project down to the model's embedding dimension.

## A.5 INPUT TO THE GREAT BASELINE

As mentioned in the main text, we also compare with GREAT Hellendoorn et al. (2020). Since their preprocessing pipeline is proprietary and could not be shared with us even after contacting the authors, we provide to GREAT the same AST distances as our model. Since GREAT uses edges instead of distances to encode relations in the Structure, we essentially threshold the ancestor, sibling, and shortest-paths distances and provide the edges where the distances are equal to 1 (including their edge types) to the model.

|       | (a) CODE TRANSFORMER |
|-------|-----|
| Hyperparameter | Value |
| Activation | GELU |
| Input Nonlinearity | tanh |
| Num. layers | 3 |
| $d$ | 1024 |
| $d_{FF}$ | 2048 |
| $p_{\text{dropout}}$ | 0.2 |
| Num. heads | 8 |

(b) GREAT (Hellendoorn et al., 2020)

| Hyperparameter | Value |
|-------|-----|
| Activation | GELU |
| Num. layers | 3 |
| $d$ | 1024 |
| $d_{FF}$ | 2048 |
| $p_{\text{dropout}}$ | 0.2 |
| Num. heads | 8 |

Table 7: Code Summarization hyperparameters

## A.6 EXPERIMENTAL SETUP

Table 7 shows hyperparameters of our models for code summarization. For all our experiments, we use a Transformer Decoder with one layer and teacher forcing to generate 6 output sub tokens. We also employ label smoothing of 0.1. As optimizer, we use Adam with a learning rate of $8e^{-5}$ and weight decay of $3e^{-5}$. Batch size during training is 8 with a simulated batch size of 128 achieved by gradient accumulation.

Apart from comparing the CODE TRANSFORMER to baselines, we performed the following hyperparameter comparisons and ablation studies:

- CODE TRANSFORMER (structure-only)
  Using only AST information as input, i.e., masking all tokens that do not correspond to a leaf of the AST, and removing the token distance as a relation to be used by the model. Further, token types are not fed into the model.

- CODE TRANSFORMER (context-only)
  Here, we do not include any information on the AST (i.e. node types and distances on the AST). This is effectively the XLNet backbone plus encoding of the token type returned by the tokenizer.

- CODE TRANSFORMER (Max-Dist.)
  Applying a Max Distance Mask of 5 to the shortest paths distance (i.e., model cannot see a node that is more than 5 hops away no matter how small the other distances are). Early results showed that, as expected, results deteriorate substantially when limiting our model's receptive field. Hence, we do not include these results in this work.

- Using 16 and 64 bins instead of 32 bins. This had no noticeable effect on performance.

## A.7 CODE SUMMARIZATION EXAMPLES

In the Tables 8, 9, 10, 11, 12, 13, 14 and 15 we present example functions from the Java-small dataset along with the different models' predictions for the function name.

```java
public Summation next() {
    return parts[i++];
}
```

| Model | Prediction |
|---|---|
| GREAT | get x map |
| code2seq | get parts |
| Ours w/o structure | get |
| CODE TRANSFORMER | get next |
| Ground Truth | next |

Table 8: The CODE TRANSFORMER is the only model to correctly identify the notion of getting the *next* entry.

```java
private Path findCacheFile(Path[] cacheFiles, String fileName) {
    if (cacheFiles != null && cacheFiles.length > 0) {
        for (Path file : cacheFiles) {
            if (file.getName().equals(fileName)) {
                return file;
            }
        }
    }
    return null;
}
```

| Model | Prediction |
|---|---|
| GREAT | get path |
| code2seq | find file |
| Ours w/o structure | get file |
| CODE TRANSFORMER | find cache |
| Ground Truth | find cache file |

Table 9: The CODE TRANSFORMER is the only model to both recognize that the task is to *find* a file as well as the fact that it is about the cache. However, it did not correctly predict the *file* part of the method name.

```java
public int compare(Pair<LoggedJob, JobTraceReader> p1,
                   Pair<LoggedJob, JobTraceReader> p2) {
    LoggedJob j1 = p1.first();
    LoggedJob j2 = p2.first();
    return (j1.getSubmitTime() < j2.getSubmitTime()) ? -1
            : (j1.getSubmitTime() == j2.getSubmitTime()) ? 0 : 1;
}
```

| Model | Prediction |
|---|---|
| GREAT | run |
| code2seq | get submit time |
| Ours w/o structure | compare |
| CODE TRANSFORMER | compare |
| Ground Truth | compare |

Table 10: The CODE TRANSFORMER and the its context-only variant are the only models correctly recognizing the 'compare' template in the method body.

```java
public static MNTPROC fromValue(int value) {
    if (value < 0 || value >= values().length) {
        return null;
    }
    return values()[value];
}
```

| Model | Prediction |
|---|---|
| GREAT | get value |
| code2seq | get value |
| Ours w/o structure | to |
| CODE TRANSFORMER | from value |
| Ground Truth | from value |

Table 11: The CODE TRANSFORMER is the only model to recognize that the snippet is similar to a static factory method which is often preceded with *from*.

```java
private Iterable<ListBlobItem> listRootBlobs(String aPrefix,
                                             boolean useFlatBlobListing,
                                             EnumSet<BlobListingDetails> listingDetails,
                                             BlobRequestOptions options,
                                             OperationContext opContext)
                                    throws StorageException, URISyntaxException {
    CloudBlobDirectoryWrapper directory = this.container.getDirectoryReference(aPrefix);
    return directory.listBlobs(null, useFlatBlobListing,
                          listingDetails, options, opContext);
}
```

| Model | Prediction |
|---|---|
| GREAT | list blobs |
| code2seq | list blobs |
| Ours w/o structure | list blobs |
| CODE TRANSFORMER | list blobs by prefix |
| Ground Truth | list root blobs |

Table 12: All models could correctly identify the *listBlobs()* call in the return statement. However, the CODE TRANSFORMER additionally comprehended that the specified prefix is quite important.

```java
private static void dumpOpCounts(EnumMap<FSEditLogOpCodes, Holder<Integer>> opCounts) {
    StringBuilder sb = new StringBuilder();
    sb.append("Summary of operations loaded from edit log:\n  ");
    Joiner.on("\n  ").withKeyValueSeparator("=").appendTo(sb, opCounts);
    FSImage.LOG.debug(sb.toString());
}
```

| Model | Prediction |
|---|---|
| GREAT | append |
| code2seq | add |
| Ours w/o structure | log |
| CODE TRANSFORMER | log op counts |
| Ground Truth | dump op counts |

Table 13: Only the CODE TRANSFORMER could correctly identify that it is the *op counts* that should be logged.

```java
static String execCommand(File f, String... cmd) throws IOException {
    String[] args = new String[cmd.length + 1];
    System.arraycopy(cmd, 0, args, 0, cmd.length);
    args[cmd.length] = f.getCanonicalPath();
    String output = Shell.execCommand(args);
    return output;
}
```

| Model | Prediction |
|---|---|
| GREAT | get canonical path |
| code2seq | exec |
| Ours w/o structure | get output |
| CODE TRANSFORMER | exec command |
| Ground Truth | exec command |

Table 14: Only the CODE TRANSFORMER and code2seq could identify that the relevant part of the method is concerned with executing a command instead of returning something.

```java
protected void subView(Class<? extends SubView> cls) {
    indent(of(ENDTAG));
    sb.setLength(0);
    out.print(sb.append('[').append(cls.getName()).append(']').toString());
    out.println();
}
```

| Model | Prediction |
|---|---|
| GREAT | print |
| code2seq | print |
| Ours w/o structure | print |
| CODE TRANSFORMER | print sub view |
| Ground Truth | sub view |

Table 15: Only the CODE TRANSFORMER was able to link the *print* functionality to the object that should be printed, which can only be inferred from the object's class in the method parameters.

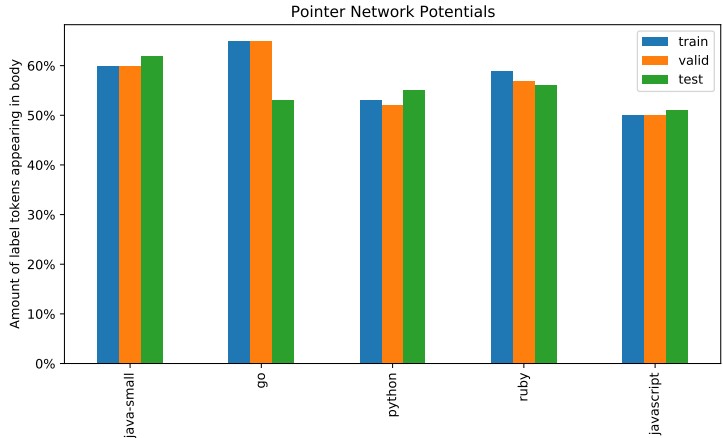

Figure 7: Share of tokens in the labels also occurring in the bodies of methods.

| Model | Python Prec. | Python Rec. | Python F1 | Javascript Prec. | Javascript Rec. | Javascript F1 | Ruby Prec. | Ruby Rec. | Ruby F1 | Go Prec. | Go Rec. | Go F1 |
|---|---|---|---|---|---|---|---|---|---|---|---|---|
| code2seq | - | - | - | - | - | - | - | - | - | - | - | - |
| GREAT | 34.93 | 31.12 | 31.61 | 29.69 | 24.24 | 25.55 | 25.69 | 21.49 | 22.18 | 48.38 | 45.97 | 45.71 |
| Ours w/o structure | 36.87 | 32.17 | 32.97 | 31.30 | 25.03 | 26.64 | 31.43 | 25.34 | 26.63 | 49.78 | 46.73 | 46.69 |
| Ours w/o pointer net | **38.77** | 31.72 | 33.27 | 32.70 | 25.50 | 27.33 | **32.12** | **30.17** | **29.36** | **53.09** | **48.70** | **49.26** |
| Ours | 36.68 | **33.86** | **33.84** | **33.36** | **27.55** | **29.02** | 31.53 | 24.72 | 26.43 | 52.00 | 47.35 | 47.93 |
| code2seq (Multilanguage) | - | - | - | - | - | - | - | - | - | - | - | - |
| GREAT (Multilanguage) | 35.73 | 30.81 | 31.74 | 31.49 | 26.17 | 27.41 | 29.72 | 24.20 | 25.43 | 50.32 | 47.94 | 47.66 |
| Ours w/o structure (Mult.) | 36.78 | 29.92 | 31.58 | 32.60 | 26.02 | 27.74 | 31.71 | 26.07 | 27.24 | 51.91 | 47.58 | 48.15 |
| Ours w/o pointer (Mult.) | 37.18 | 30.52 | 32.04 | 33.95 | 25.92 | 28.11 | 32.76 | 25.04 | 27.01 | 53.50 | 48.54 | 49.35 |
| Ours (Multilanguage) | **38.10** | **33.32** | **34.18** | **34.29** | **28.69** | **30.08** | **33.30** | **28.33** | **29.29** | **53.86** | **50.46** | **50.61** |
| Ours (Mult. + Finetune) | 38.29 | 32.41 | 33.65 | 34.43 | 28.28 | 29.91 | 32.89 | 27.15 | 28.49 | 53.85 | 50.85 | 50.81 |
| Ours (Mult. + LM Pretrain) | **38.97** | **34.77** | **35.34** | **35.23** | **30.26** | **31.38** | **33.73** | **29.15** | **29.94** | **55.31** | **52.03** | **52.13** |

Table 16: Code summarization results on the CSN dataset (sample-F1).

## A.8 ESTIMATION OF POINTER NETWORK POTENTIAL

In Table 2 we observe that the pointer network improves the F1 score for all languages except Go, where counterintuitively it leads to reduced performance as measured by F1 score on the test set (while it improves by about 3 points on validation). To investigate this, in Figure 7 we plot the share of tokens in the labels also occurring in the bodies of methods in the different languages. Intuitively, this gives an indication on how much gain we can expect from using a pointer network. If the share were zero, then *no* token in the labels ever occur in the bodies of the methods, so the pointer network cannot improve the prediction by pointing at the input. We see that for Go, there is a strong mismatch between the test partition and the train/validation partitions, which much fewer tokens from the labels occurring in the bodies of methods on test compared to train/validation. Thus, we attribute the drop in performance observed by adding a pointer network on Go to this apparent violation of the i.i.d. assumption.

## A.9 CODE SUMMARIZATION RESULTS ON THE CSN DATASET (SAMPLE-F1)

In Table 16, we present our results on the CSN dataset as measured by the sample-F1 score.

