# OpenReview forum: "Language-Agnostic Representation Learning of Source Code from Structure and Context"
_ICLR.cc/2021/Conference — ICLR 2021 Poster_

### Official Review · AnonReviewer1 · 2020-10-19

**Rating:** 6
**Confidence:** 5

**Review:**

#### Summary
This paper presents a representation of source code based on the AST. By adding positional relational information, such as shortest path length, and ancestor distance the transformers learn to better represent code, without language specific features. On the code summarization task, the model improves on baselines, while training among many languages further improves results. The representations learned seem to share semantic similarities among languages.

#### Overview
(+) State-of-the-art results for single/multiple language method naming task.

(-) Limited novelty on the machine learning modelling (paper adopts a pre-existing formulation of relational transformers from Dai _et al._ (2019) and Yang _et al._ (2019)).

(-) Only one task evaluated. Does this method offer a language-agnostic representation that generalizes to other tasks on source code?

~~(-) Some additional evaluation would be useful to discern how the proposed model and others compare.~~ [Addressed at response]

#### Comments
* It is not clear to which extent the representation used here is language agnostic vs. the model is able to learn language-agnostic features: GitHub Semantic (the package used to extract the ASTs in this work) goes into a lot of effort to convert code into a language-agnostic AST format (for its downstream analyses). In that sense, the input AST is mostly language-agnostic, which may let any neural model to be language-agnostic.
This suggests that other models that could accept Semantic's ASTs (e.g. [a], [Alon 2019a], [Fernandes 2019]) could also act as language-agnostic models, however, none of these are evaluated in the multilingual setting.
* An ablation to study the effects of the four different relations considered (Section 3.1) would be useful to understand how much they improve the model (shortest path length, vs. ancestor distance vs. PPR vs. some combinations)


#### Clarification Questions
Finally, I have a few clarification questions (which I think need to also be clarified in the paper):
* Are the relational encodings an input on all transformer layers or just the first one (like the "standard" BERT positional encoding)?
* The "context" has the "token distance" is this the absolute (BERT-like) or the relative (like Dai _et al._ 2019, Yang _et al._ 2019)
* In 5.2 it's unclear what is the representation of each function. Is it the representation of the masked method name token? (the "encoder output of the method name" is a bit vague, but suggests that)
* Are all AST nodes input elements to the "Structure+Context" model?
* How does GREAT differ from the "context" model? Is it just the relational transformer (scalar bias) formulation, the 1-step AST relations and the absolute positional encodings?

#### Additional References
[a] Kim, Seohyun, et al. "Code Prediction by Feeding Trees to Transformers." arXiv preprint arXiv:2003.13848 (2020).

---

> ### Author Response · Authors · 2020-11-20
> **Authors' response to Reviewer 1**
>
> We thank Reviewer 1 for their thoughtful and constructive feedback. Below we address the points and questions raised by the reviewer.
> * Regarding the concern of limited novelty of our method, we would like to highlight that, while our main contribution is not in proposing a brand new Transformer architecture, our model is able to leverage multiple distances, as well as distances with continuous values (e.g. PPR), which is not possible with the mentioned base models.
> * As common in recent ML4Code papers, we opt to study one popular task in-depth as opposed to a breadth-oriented study of many different tasks. That being said, we do not make any task-specific modeling choices, so we believe that our proposed model is also able to perform well on other tasks; such exploration is a promising direction for future work.
> * To address the point of better comparison of our model and the baselines, we performed the following additional experiments. (i) To better compare with the code2seq baseline on the CSN dataset, we additionally train our model without a pointer network on the individual languages as well as in the multi-language setting. The results -- added to Table 2 in the updated manuscript -- show that in this setting our model still outperforms code2seq by a large margin. (ii) We additionally trained GREAT in the multi-language setting (see Table 2 and Section 5.2 in the revised manuscript). We find that, similar to our model, GREAT also benefits from the multi-language training (yet is outperformed by our multi-language model), highlighting that multi-language training is a promising direction for any ML model on code.
> * To the point of language-agnostic representations, we agree with the reviewer that the multi-language setting is facilitated by the GitHub Semantic tool. Further, as indicated in our previous point, our results suggest that any Structure-based model could benefit from multi-language training; though the improvement was strongest for our model. This opens exciting directions for future research.
> * Based on the reviewer’s request for an ablation study of the different AST-based distance functions, we performed the following additional experiment. For each of the four different AST-based distances, we train our model using only this individual distance function in the structure-only setting (without pointer network) on Java-small. We have included this additional ablation study in Table 4 of the updated manuscript and added a paragraph in Section 5.1. In summary, while the personalized PageRank distance performs best on its own,  we find that each of the individual distances performs substantially worse than their combination, highlighting the usefulness of combining the distances in our model as well as their complementary nature.
>
> In the following, we address the reviewer’s clarification questions.
>
> Q: Are the relational encodings an input on all transformer layers or just the first one (like the "standard" BERT positional encoding)?
> A: The (relative) positional encodings are used at each layer of the model (as proposed by Dai et al. (2019)).
>
> Q: The "context" has the "token distance" is this the absolute (BERT-like) or the relative (like Dai et al. 2019, Yang et al. 2019)
> A: The context distance (i.e. sequence distance) also uses relative positions, i.e. the relative position of the token pairs in the sequence like in Dai et al. 2019, Yang et al. 2019.
>
> Q: In 5.2 it's unclear what is the representation of each function. Is it the representation of the masked method name token?
> A: The reviewer is correct; we use the representation of the masked method name at the final layer of the encoder as the representations for the snippet. We have updated our description in Section 5.2 to improve clarity.
>
> Q: Are all AST nodes input elements to the “Structure+Context” model?
> A: Importantly, our model integrates both types of information, which allows it to have strong empirical performance. In particular, we map sequence tokens to AST nodes. That is, a token is assigned to the AST node whose source span is the shortest to contain it. These tokens’ input is concatenated with the assigned AST node’s type. Among all the internal nodes, we use as input only those corresponding to a token in the sequence. However, the remaining nodes are considered by the model, since their presence affects the AST distances of the remaining nodes in the AST. We have added a paragraph on this to Section 3.1 in the revised manuscript.
>
> Q: How does GREAT differ from the "context" model? Is it just the relational transformer (scalar bias) formulation, the 1-step AST relations and the absolute positional encodings?
> A: Regarding the relation of GREAT to our Context-only model (i.e. no AST information), the reviewer’s assessment is correct: the main differences are that GREAT uses absolute position encodings and leverages AST information in the form of 1-hop edges on the AST in the “scalar bias” way.

---

> > ### Comment · AnonReviewer1 · 2020-11-24
> > **Thanks**
> >
> > Thanks for your response and the updates on the paper. This addresses some of my concerns and clarifies my questions. I have updated the review accordingly.
> >
> > I think that it's up to the AC to decide if the two pending concerns are important or not. Specifically,
> >
> > * If the evaluation on a single task is sufficient for evaluating the representation contribution of this paper.
> > * If the delta of the contributions of this work to those of Dai et al. (2019) and Yang et al. (2019), are sufficient for this paper to get it accepted.

---

### Official Review · AnonReviewer4 · 2020-10-27
**Interesting design, need better experimental comparison**

**Rating:** 6
**Confidence:** 4

**Review:**

This paper wants to combine sequence (called Context) and AST (called Structure) representations of source code in a Transformer encoder model. For this, it makes clever use of relative position embedding of Transformer-XL. Different pairwise relations based on ASTs and sequence ordering are computed and each of them is encoded as a separate distance matrix, with its own learnable projection. This model is evaluated on the task of code summarization and compared against code2seq and GREAT models, and against different configurations of the proposed model, called Code Transformer. The results show that Code Transformer achieves results better than these models. This evaluation is conducted on five languages. A separate evaluation with the model trained on all the languages together is also performed. This multi-lingual model outperforms the mono-lingual models.

**Design**
The paper considers four types of paths in the ASTs and quantifies the path lengths and directions. The ancestor and sibling distances are also defined over internal (non-terminal) nodes of the AST. But the input to the model is a sequence of sub-tokens (grouped by tokens). As discussed in appendix A.2 and A.3, the leaf nodes are mapped to some internal nodes whose types are concatenated to the token representations. Thus, the internal nodes themselves are not part of the input. From section 3.1, it appears as if all the internal nodes will be somehow encoded. This does not seem to be the case and the non-leaf-to-leaf paths are perhaps projected only one certain internal nodes. Please clarify this.

A separate projection matrix $W_r^{(s)}$ is used for each relation r. Is the positional encoding $\phi$ shared between them?

The paper calls the proposed model as Code Transformer. Technically, it is only the encoder. It would be good to make this clear.

In the appendix, the paper states that a token span may have overlap among its ancestors. A concrete example would help.

**Language-agnostic representation**
I find the use of the term "language-agnostic representation" confusing. The proposed method uses features computed from ASTs which are language dependent and even dependent on the parser. If a different parser for the same language is used, the features would change (because even if the grammars might be equivalent they may not be identical). I think the authors want to emphasize that they don't use program analysis information such as control and data flow. It would be more accurate to state that as such.

**Learned representations**
The paper presents nice visualizations, and examples of methods and their embeddings. However, it is important to highlight in the paper that these representations are learned in the context of a specific supervised task, unlike unsupervised representations, eg, in "Learning and evaluating contextual embedding of source code" (ICML'20). Note that those can also be used for code summarization with a decoder similar to this paper or by using other decoding methods such as "Blank language models" (https://arxiv.org/abs/2002.03079).

The paper talks about pre-training on Java-large and Java-medium datasets and fine-tuning on Java-small. What is the pre-training objective? If masking (similar to BERT) is involved, then how are the distance matrices computed and masked?

**Baselines**
The GREAT model uses semantic and syntactic relations between tokens (called leaves-only graph). However, this paper uses the ASTs, which is different from the leaves-only representation. First, how exactly the ASTs are used with GREAT is not explained. Second, the results do not imply that using Structure, as defined in this paper, can outperform edge-level representation and relational attention for the edge types considered in the GREAT paper.

Another baseline considered is Fernandes et al. (2019) on the Java-small dataset. The performance improvement over this baseline is marginal (F1 of 51.4 vs 51.83 with pre-training).

---

> ### Author Response · Authors · 2020-11-20
> **Authors' response to Reviewer 4**
>
> We thank Reviewer 4 for their thoughtful and constructive feedback. In the following, we address the points raised by the reviewer:
> * Regarding the mapping from leaf nodes to internal nodes: among all the internal nodes, we use as input only those corresponding to a token in the sequence. However, even nodes which have no token assigned to them are considered by the model, since their presence affects the AST distances of the remaining nodes in the AST. We have added a paragraph to explain this to Section 3.1 in the revised manuscript.
> * Regarding the second question about the encoding: we use the standard sinusoidal encoding function proposed by Vaswani et al. 2017, which is the same for all relations and contains no learnable parameters. We have added a sentence to Section 3 as well as the definition of the encoding functions to Appendix A.1 in the revised manuscript.
> * We refer to our full model as the “Code Transformer” since it also contains a Transformer decoder equipped with a pointer network; when we refer to the encoder specifically (e.g., when studying the multilingual representations), we updated the text in order to make it explicit.
> * Regarding the overlap of token spans mentioned in the appendix, we have included a minimal example when this happens in Fig. 6 in the appendix. Basically, when assigning the token “NoReturn” to an AST node, we start from the root and recursively expand the child node whose span contains the token; in the rare case where both children contain the token under consideration (as in the example in Fig. 6), we employ a heuristic that selects the child node whose span is smaller. In the example, this selects the correct node and the “NoReturn” token is assigned to the “Identifier” child of the “Annotation” node. Note that in our experiments, this occured only in about 1 in 1000 code snippets.
> * Regarding the term “language-agnostic” used to describe our approach, as the reviewer correctly states, we want to highlight that by not relying on any language-specific features (such as program analysis edges) our model can be trained in a multi-language fashion, which substantially improves performance. This is an excellent point and a real strength of our approach. While we agree that GitHub Semantic facilitates our multi-language setting by providing unified ASTs for different languages, we argue that the term “language-agnostic” is justified since nowhere in our model we make language-specific choices. Further, as indicated by our additional experiments using GREAT in the multi-language setting (see Table 2 in the revised manuscript), our results suggest that any Structure-based neural model could benefit from multi-language training; though the improvement was strongest for our model. This opens exciting directions for future research.
> * We thank the reviewer for providing the interesting reference to "Learning and evaluating contextual embedding of source code" (ICML'20); we have included a reference to Section 5.2 of the updated manuscript, explaining how our learned embeddings relate to the ICML’20 work (i.e., our embeddings are task-specific and multilingual vs. task-agnostic and monolingual in the ICML’20 work). In addition, we highlight that it is also possible to use our model to obtain multilingual embeddings in a downstream-task-agnostic way using our model, i.e., when pre-training on the masked language-modeling (MLM) objective.
> * Based on the reviewer’s suggestion, we added a paragraph explaining how we use the AST information in GREAT to Appendix A.5 in the revised manuscript.
> * The pre-training objective we employ for our model is the permutation language modeling task proposed by XLNet (Yang et al. 2019), which is similar to BERT’s masked language modeling yet does not require [MASK] tokens. In our model, we mask the input tokens’ contents (including their corresponding AST node types), but not their position in the snippet (i.e., AST distances and token sequence distance). We have added a clarification to Section 5.1 of the revised manuscript.
> * Our main goals are to show that (i) relying only on structure features that can easily be computed for any programming language brings strong advantages since it enables multi-language training; (ii) our formulation of using the structure information as relative distances brings advantages over localized, edge-based structure information as used in GREAT. Note that it is possible to extend our model to use program analysis edges, so comparing GREAT and our Code Transformer in this setting is an interesting aspect to study in future work.
> * Regarding the improvement in code summarization on Java-small, we would like to highlight that the improvement mentioned by the reviewer (51.83 vs 51.4) is without pre-training; when pre-training our model first, we achieve an F1 score of 53.77.

---

> > ### Comment · AnonReviewer4 · 2020-11-24
> > **Better explanations**
> >
> > Thank you for clarifying my doubts and updating the paper. It explains how the inputs are constructed in terms of ASTs.

---

### Official Review · AnonReviewer3 · 2020-10-28
**Simple extension of transformers for code that works well.**

**Rating:** 7
**Confidence:** 4

**Review:**

There is not much not to like about this paper as it has a simple idea to extend the transformer model. The paper proposed to not only take positional information of each token, but to add additional structural information about distances of tokens in the abstract syntax tree. This positional information gives an edge of this model on several code summarization datasets. The simplicity of the proposed model (assuming it is released by the authors) puts the work in the state-of-the-art category in machine learning for code.

The paper is also easy to follow and the contribution, while small, is clear and well explained. The related work is also thoroughly covered. While it looks standard, the idea seems to deliver well in the evaluation results. This is also the first work that shows cross-language improvements for programming language models.

In terms of writing, my main complaint is that the paper can better relate to existing works. For example, “ours without structure” seems to be equivalent to a “plain” transformer with a pointer network in the decoders and using relative distances. It would also help if Table 2 includes a row without the pointer network called a transformer model.

In the light of this, it also looks like the comparison with the GREAT model is not quite fair. One possibility to fix this is to encode some program analysis edges at least for one language. The program analysis edges are in fact not difficult to build, especially if the authors want to claim they did this comparison correctly.

As another possible improvement, I find the terms “structure” and “context” confusing and I would suggest the authors use something that directly refers to trees and sequences.

---

> ### Author Response · Authors · 2020-11-20
> **Authors' response to Reviewer 3**
>
> We thank Reviewer 3 for their thoughtful and constructive feedback. In the following, we address the points raised by the reviewer:
> * While the reviewer is correct that “ours without structure” is related to an XLNet Transformer (with pointer network), we would like to highlight one key difference. “Ours without structure” also takes as input the token type (as returned by the tokenizer) of each token, an adaptation to deal with code. Moreover, we refer to the model “Ours without structure” in the tables in order to make the ablation (i.e., how it relates to our full model) clear to the reader. We added a sentence on this to Section 5.1 in the revised manuscript.
> * Following the reviewer’s suggestion, we trained our model without pointer network on CSN. The results can be found in Table 2 of the revised manuscript. In summary, we observe that the pointer network helps in all but one case. The one case where the model without a pointer network actually performs better is single-language training on Go, while on the validation set the variant with pointer network is ahead by 3 points. To investigate this further, we inspected the datasets and computed an upper-bound of how much performance improvement can be expected from the pointer network, i.e., we computed the share of label subtokens that also occur in the input sequence (a high fraction indicates that the pointer network can contribute substantially to the performance). As displayed in Figure 7 (appendix A.8) of the revised manuscript, in the Go dataset we observe that the fraction described above is much higher on the training and validation set compared to the test set. We hypothesize that because of this violation of the i.i.d. assumption in the split of the dataset creators, the pointer network actually hurts the performance on Go. We included these results in the appendix for completeness.
> * Regarding GREAT, we believe that our comparison is fair in the sense that GREAT has access to the same information as our model. Besides the fact that their preprocessing pipeline is not public and could not be shared with us (even after contacting the authors), including program analysis edges brings several disadvantages: (i) it means that we would require a custom pipeline and representation for each of the languages, making it impractical for the multi-language setting we highlight in this work; (ii) extracting program analysis edges is not possible for purely interpreted languages such as Python or Javascript, thus limiting the generality of the model. In fact, if the goal was to focus only on specific (compiled) languages and to use as much language-specific domain knowledge as possible, it would be straightforward to extend the Code Transformer to include program analysis edges, and compare to GREAT in that setting. Although this setting is out of scope for the present paper, we believe it represents a promising avenue for future work. Additionally, we trained GREAT in the multi-language setting (see Table 2 and Section 5.2 in the revised manuscript) for comparison. We find that, similar to our model, GREAT also benefits from the multi-language training (yet is outperformed by our multi-language model), highlighting that multi-language training is a promising direction for any representation learning model on code.
> * We appreciate the reviewer's feedback regarding the terms "Structure" and "Context". As such, we made sure to clarify their meaning across the paper in order to remove any ambiguity for the reader.

---

> > ### Comment · AnonReviewer3 · 2020-11-24
> > **Thank you**
> >
> > Thank you for the explanations. I appreciate the fixed. The only small remaining remark is that it is still not correct to call a baseline system to be the GREAT algorithm if it doesn't use the same (or at least reimplemented similar) algorithm. It can be called transformer or anything else, but it is not the same as it was not implemented. I understand that the authors should not spend time to replicate works that are not available, but then they may simply not cite that algorithm in the results.

---

### Official Review · AnonReviewer2 · 2020-11-04
**Review of "Language-Agnostic Representation Learning of Source Code from Structure and Context"**

**Rating:** 7
**Confidence:** 4

**Review:**

Summary:

The authors develop a Transformer model for language-agnostic code summarization. The Transformer is provided both sequential tokenized code and a parsed AST as inputs, and tasked with generating the name of the corresponding code function as output. The authors show that the inclusion of this added structural AST information improves performance on the task, and also improves the cross-language transfer learning abilities of the model. They demonstrate state-of-the-art performance on this task when testing against comparable architectures and datasets. This reviewer believes that the paper is deserving of acceptance to ICLR 2021.

Reasons to Accept:
- The authors take pains to fairly compare their model to prior works, including testing on two separate datasets, and ablating their Pointer Network decoder.
- The authors achieve state-of-the-art results for the code-summarization task on both the CodeSearchNet dataset (Python, Javascript, Ruby, Go) and the Java-small dataset.
- The authors demonstrate the surprising extent to which structural information can help with cross-language generalization, particularly for undersampled languages.
- The authors visualize the embeddings learned by the Transformer, and compare them across languages.

General Issues to resolve:
- Given the centrality of AST code representations to this paper, there should be some figure showing a sample code snippet alongside its AST in full. This can go in the appendix if need be.
- Clarify how Ancestor (and Sibling) distance are defined when one node is not a direct ancestor (or sibling) of another node.
- Add a more detailed explanation to Figure 3, particularly regarding the "Attention Transformer" subfigure.

Grammar/Syntax Issues:
- Bottom of Page 2: "learn structure (syntax) and *semantics* from scratch"
- Bottom of Page 4: $r_{v_1 \to v_3} = -2$  should be  $r_{v_3 \to v_1} = -2$
- Bottom of Page 4: "PPR is a well-studied proximity measure *which* has"
- Middle of Page 13: ['set', 'bottom', 'height'] should be rendered using backticks
- Middle of Page 13: '\n' should be rendered using backticks
- Middle of Page 15: "the child *has* a distance"
- Numbering of Tables goes from 6 to 8, skipping 7

---

> ### Author Response · Authors · 2020-11-20
> **Authors' response to Reviewer 2**
>
> We thank Reviewer 2 for their thoughtful and constructive feedback. We are grateful for the grammar / syntax errors spotted by the reviewer, and note that they have been fixed in the revised manuscript. In the following, we explain how we have addressed the reviewer’s feedback in the revised version:
> * We have included an example snippet and its corresponding AST  (Fig. 6 in the appendix).
> * When node B is not a direct sibling or ancestor of the considered node A, we consider B to be unreachable from A by following only sibling or ancestor edges, respectively. We encode this “unreachability” by using a very large value (e.g., 1000) for the respective distance between nodes A and B.  Using the sinusoidal distance encoding function, the model can thus effectively distinguish between unreachable nodes and the siblings / ancestors / etc. of node A. We have extended our description in Appendix A.3.3 in the revised manuscript to make this clear.
> * Based on the reviewer’s suggestion, we extended the caption of Figure 3 in the updated manuscript.

---

### Public Comment · ~Uri_Alon1 · 2020-11-15
**Questions about pretraining**

Hi,
Thank you for this paper, this is a really nice paper with a very extensive evaluation.
The multilingual experiments are a promising direction and this is the first paper to train multilingual models and show their benefits (as far as I know).

I have a question about pretraining on Java-large and Java-medium and fine-tuning on Java-small -

1. Did you try to pre-train any of the baselines on the large-medium datasets as well? It would be interesting to see the relative gain from pre-training of your Code Transformer compared to the baselines.
2. Why do you need to fine-tune on Java-small? The training sets of Java-large and Java-small come from the same domain, so I even expect that pretraining on {large,medium} *without* fine-tuning on Java-small would work better, because there is no need to bias the model toward the training set of Java-small specifically.

An additional possible reference -
"Novel positional encodings to enable tree-based transformers", Shiv & Quirk, NeurIPS 2019.
Feel free to ignore the following question, I am just sharing my thoughts, I know that it is open-ended -
I wonder if their tree-based positional encodings are more/less expressive, and more/less generalizable than the set of pairwise relations described in Section 3.1?

Thanks!
Uri

---

> ### Author Response · Authors · 2020-11-18
> **Authors' response**
>
> Dear Uri,
>
> Thanks for your interest in our work! We are really grateful for your encouraging feedback, especially considering your expertise as the main author of code2seq. In the following, we address your questions in detail.
>
> 1. Pre-training the baselines on Java-{medium | large} before fine-tuning on Java-small is an interesting direction. While this is relatively straightforward to implement for GREAT and the Fernandes et al. (2019) baseline, for code2seq we would have to first come up with a pre-training task on the Structure that is equivalent to the masked language modeling task on the Context representation. While we have not tried the baselines in the pre-training setting on Java-small yet, we want to highlight that in the revised manuscript (available in a few days) we will include results for GREAT in the multi-language setting on the CSN dataset, where we observe that GREAT also benefits from the multi-language training. We believe that this is an exciting finding, suggesting that any model that relies on the Structure could benefit from the multi-language setup outlined in our paper.
> 2. We perform fine-tuning on Java-small because our pre-training task is masked language modeling, and therefore the pre-trained model is not yet able to perform well on code summarization. We argue that the pre-training task should be agnostic to the fine-tuning tasks for which the model is used later. Of course we could also first train the model on the downstream task (i.e., code summarization) on the full Java-{medium | large} dataset, but then we argue that fine-tuning on Java-small would not be meaningful – we could simply evaluate the model on the large dataset instead.
>
> Thank you also for the reference on positional encodings on trees. We will include a reference to this work in the revised manuscript, which we will upload in a few days (including also all our responses to the reviewers). While the proposed approach looks very interesting, one major limitation is that it assumes regular trees, which makes it impractical in the case of ASTs, since for example methods (nodes) can have arbitrarily many input arguments (children). However, we believe that adapting their approach to support arbitrary trees is a promising direction for future research.
>
> Best,
>
> The authors

---

### Decision · Program_Chairs · 2021-01-07
**Final Decision**

**Decision:**

Accept (Poster)

**Comment:**

The paper gives an extension of the transformer model that is suited to computing representations of source code. The main difference from transformers is that the model takes in a program's abstract syntax tree (AST) in addition to its sequence representation, and utilizes several pairwise distance measures between AST nodes in the self-attention operation. The model is evaluated on the task of code summarization for 5 different languages and shown to beat two state-of-the-art models. One interesting observation is that a model trained on data from all languages outperforms the monolingual version of the model.

The reviewers generally liked the paper. The technical idea is simple, but the evaluation is substantial and makes a convincing case about setting a new state of the art. The observation about multilingual models is also interesting. While there were a few concerns, many of these were addressed in the authors' responses, and the ones that remain seem minor. Given this, I am recommending acceptance as a poster. Please incorporate the reviewers' comments in the final version.